# Seismic evidence for a cold serpentinized mantle wedge beneath Mount St Helens

S.M. Hansen[1], B. Schmandt[1], A. Levander[2], E. Kiser[2,†], J.E. Vidale[3], G.A. Abers[4] & K.C. Creager[3]

Mount St Helens is the most active volcano within the Cascade arc; however, its location is unusual because it lies 50 km west of the main axis of arc volcanism. Subduction zone thermal models indicate that the down-going slab is decoupled from the overriding mantle wedge beneath the forearc, resulting in a cold mantle wedge that is unlikely to generate melt. Consequently, the forearc location of Mount St Helens raises questions regarding the extent of the cold mantle wedge and the source region of melts that are responsible for volcanism. Here using, high-resolution active-source seismic data, we show that Mount St Helens sits atop a sharp lateral boundary in Moho reflectivity. Weak-to-absent PmP reflections to the west are attributed to serpentinite in the mantle-wedge, which requires a cold hydrated mantle wedge beneath Mount St Helens ($< \sim 700\,°C$). These results suggest that the melt source region lies east towards Mount Adams.

[1] Department of Earth and Planetary Sciences, MSCO3-2040, 1 University of New Mexico, Albuquerque, New Mexico 87131-0001, USA. [2] Department of Earth Science, MS-126 Rice University, 6100 Main Street, Houston, Texas 77005, USA. [3] Department of Earth and Space Sciences, University of Washington, Johnson Hall Rm-070, Box 351310, 4000 15th Avenue NE, Seattle, Washington 98195-1310, USA. [4] Department of Earth and Atmospheric Sciences, Cornell University, 1103 Bradfield Hall, Ithaca, New York 14853-1901, USA. † Present address: Department of Geosciences, The University of Arizona, 1040 E. 4th Street, Tucson, Arizona 85721, USA. Correspondence and requests for materials should be addressed to S.M.H. (email: stevehansen@unm.edu).

Mount St Helens is widely recognized as a hazardous volcano due to its high level of activity, as well as its proximity to populated regions[1]. Following the Plinian eruption that occurred on May 18, 1980, a great deal has been learned about the shallow magmatic system at Mount St Helens. Petrologic and geophysical studies indicate that most of the eruptive products were derived from one or more upper-crustal magma chambers located at ~3–12 km depth[2,3]. Arc magmas are the end result of complex processes involving the interaction between the overlying crust[4,5] and melts that buoyantly ascend from the mantle wedge source region[6]. Despite the genetic link between subduction and arc volcanism, the structure of the deep magmatic plumbing system beneath St Helens and its context within the broader Cascadia subduction system remain poorly resolved.

Mount St Helens is located in southern Washington along a portion of the Cascade Arc that displays an anomalously wide east-to-west distribution of volcanism[7] (Fig. 1). The Rainier to Hood segment contrasts sharply with the adjacent Garibaldi Volcanic Belt, which extends northward into southern British Columbia, and the central Oregon Cascade Arc, both of which define a relatively narrow axis of volcanism which is approximately parallel to the subduction trench[7] (Fig. 1). The breadth of arc volcanism in this region is exemplified by

Mount St Helens, which lies 54 km trenchward (west) of Mount Adams (Fig. 1). Beneath the forearc, the subducting Juan de Fuca plate is thought to be decoupled from the overriding mantle wedge to depths of 70–80 km resulting in a cold stagnant mantle wedge[8]. The presence of a cold forearc wedge precludes the possibility of hydrous flux melting and limits the trenchward extent of arc volcanism[6,8]. Given Mount St Helens' close proximity to the forearc region, it is unclear how arc derived melts reach the surface at this volcano.

A large-scale active-source seismic experiment was conducted at Mount St Helens during the summer of 2014 as part of the ongoing multidisciplinary imaging Magma Under St Helens (iMUSH) project. A primary goal of this project is to constrain the deep structure of the crust and mantle below Mount St Helens which has been difficult to resolve in previous seismic studies[3]. The active-source experiment consisted of 23 explosion sources that were recorded by two arrays of vertical-component geophones deployed in two phases which together contained about 4,800 individual channels[9] (Fig. 2). A Nodal array was also deployed within 15 km of the Mount St Helens summit crater and consisted of 900 autonomous seismographs containing 10 Hz geophones[10]. All sensors were deployed along the road and trail system at Mount St Helens with an average spacing of ~250 m. The resulting dataset provides a unique opportunity for high-resolution seismic imaging of deep crustal structure beneath this active arc volcano.

In this study, we use seismic energy reflected from the crust-mantle boundary (the PmP phase) to investigate spatial variations of the velocity contrast across the Mohorovičić discontinuity (Moho). The amplitude of the PmP phase is markedly reduced west of Mount St Helens and we attribute the weaker Moho velocity contrast in this region to the presence of low velocity serpentine within the mantle wedge. The presence of serpentine requires a cold and hydrated mantle wedge[11,12] which is consistent with previous geophysical studies in other regions of the Cascadia forearc[13–16] (Fig. 1). These observations suggest that

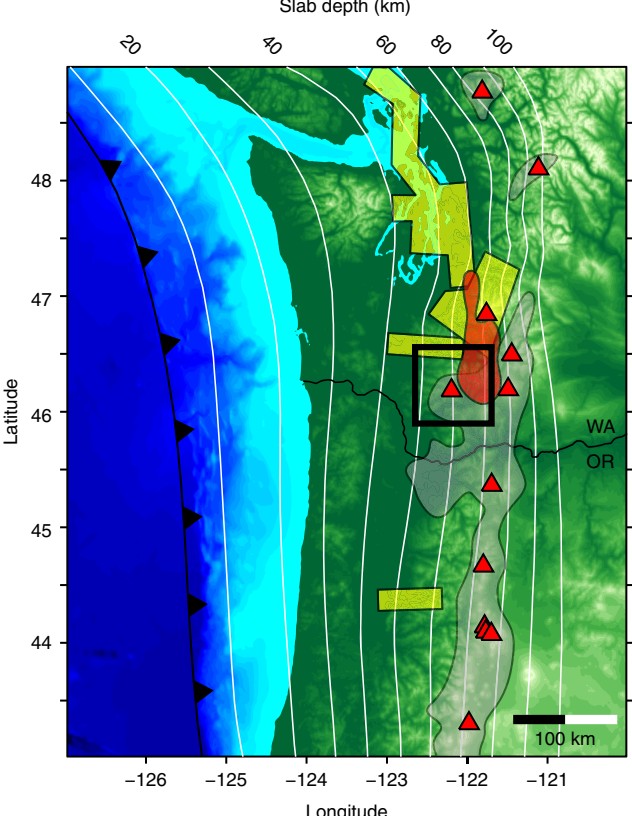

**Figure 1 | Topography and geologic features of the Quaternary Cascadia Volcanic Arc.** The location of the subduction trench is shown in black and the depth of the Juan de Fuca slab is contoured with white lines[32]. Locations of the major Quaternary volcanoes active are plotted as red triangles and the grey regions depict the distribution of volcanic vents[7]. Yellow boxes show where previous seismic experiments have observed a weak Moho arrival[13,14,18]. The red shaded region denotes the location of the Southern Washington Cascades Conductor[28,29]. The black box outlines the location of the Mount St Helens map shown in Fig. 2. The thin black lines show the state boundary between Washington (WA) and Oregon (OR).

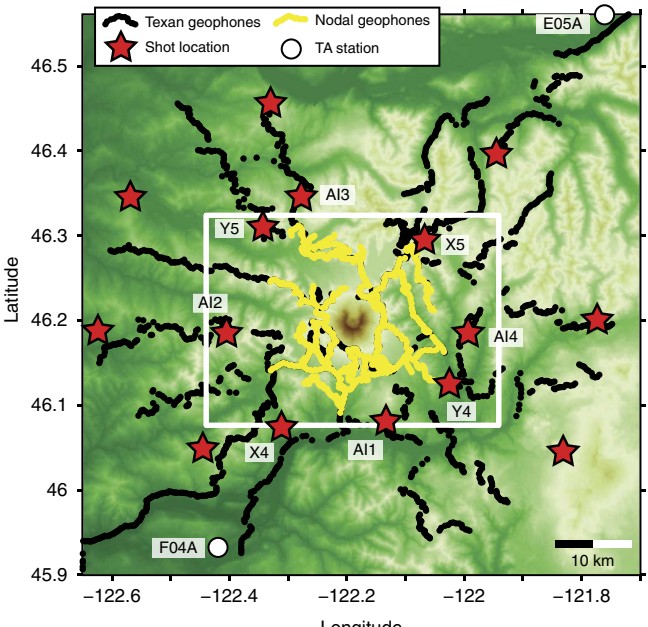

**Figure 2 | The iMUSH active-source experiment.** Seismic instrument locations are plotted as dots and shots locations are red stars. The location of the Moho amplitude map (Fig. 5) is shown as a white box. The names of shots referenced in Figs 3 and 4 are labelled. The locations of the two nearest Transportable Array (TA) stations, E05A and F04A, are also shown.

the arc magmas erupted at Mount St Helens are generated closer to the hotter axial region of the arc and migrate laterally into the cooler forearc either through the crust or upper-mantle.

## Results

**PmP amplitude variations**. The first-order observation from the recorded shot data is that the amplitude of the PmP phase changes dramatically from east-to-west across Mount St Helens. Shots occurring on the eastern side of the volcano display a strong PmP phase in both common offset and common source stacks, while shots from the west show little or no PmP energy (Fig. 3). There are no systematic differences between the size or coupling of the eastern and western shots (Fig. 4), which indicates that the observed variations in PmP amplitude are the result of lateral differences in the reflectivity of the crust-mantle boundary. The map of Moho reflectivity derived from PmP imaging (Fig. 5) shows that the boundary between the high and low reflectivity regions passes directly beneath Mount St Helens and is approximately linear with a north-northwest trend.

The amplitude of the reflected PmP phase scales with the magnitude of the velocity contrast across the crust-mantle boundary. Weak PmP amplitudes are therefore attributable to a small velocity difference across the Moho discontinuity. This implies that the velocity of the lowermost crust is similar to that of the uppermost mantle beneath the western part of the array. It is important to note that these data do not directly constrain the cause of the weak velocity contrast and that the observed westward reduction in PmP amplitudes could be caused by either an increase in lower crustal velocity, or a reduction in the velocity of the uppermost mantle relative to the eastern part of the array.

**Constraints on seismic velocity**. Initial refraction tomography efforts from the iMUSH experiment[9] indicate average lower crustal P-wave velocities of $\sim 7$–$7.7 \, \mathrm{km \, s^{-1}}$ beneath Mount St Helens, consistent with results from the 1995 wide-angle seismic experiment just north of the current study area[17]. Neither model displays velocity variations in the lower crust that can explain a systematic east-to-west reduction of PmP amplitude centred on Mount St Helens. Additional seismic constraints are provided by the two Earthscope Transportable Array stations that were nearest to the study area (Fig. 2). S-wave velocity models derived from the one-dimensional (1D) inversion of passive-source seismic data[16] (Fig. 5) support our PmP observations and show a large reduction in Moho velocity contrast from northeast to southwest across Mount St Helens. In these models, the Moho velocity step is an explicit part of the model parameterization and is primarily constrained by the amplitude of the Pms phase identified by receiver function analysis[16]. Lower crustal S-wave velocities range $\sim 3.6$–$3.9 \, \mathrm{km \, s^{-1}}$ and are only slightly higher to the southwest. The models show that the reduced Moho velocity contrast to the southwest is due to lower velocities in the uppermost mantle (Fig. 5). Taken together, the active and passive-source tomography results indicate that lateral differences in uppermost

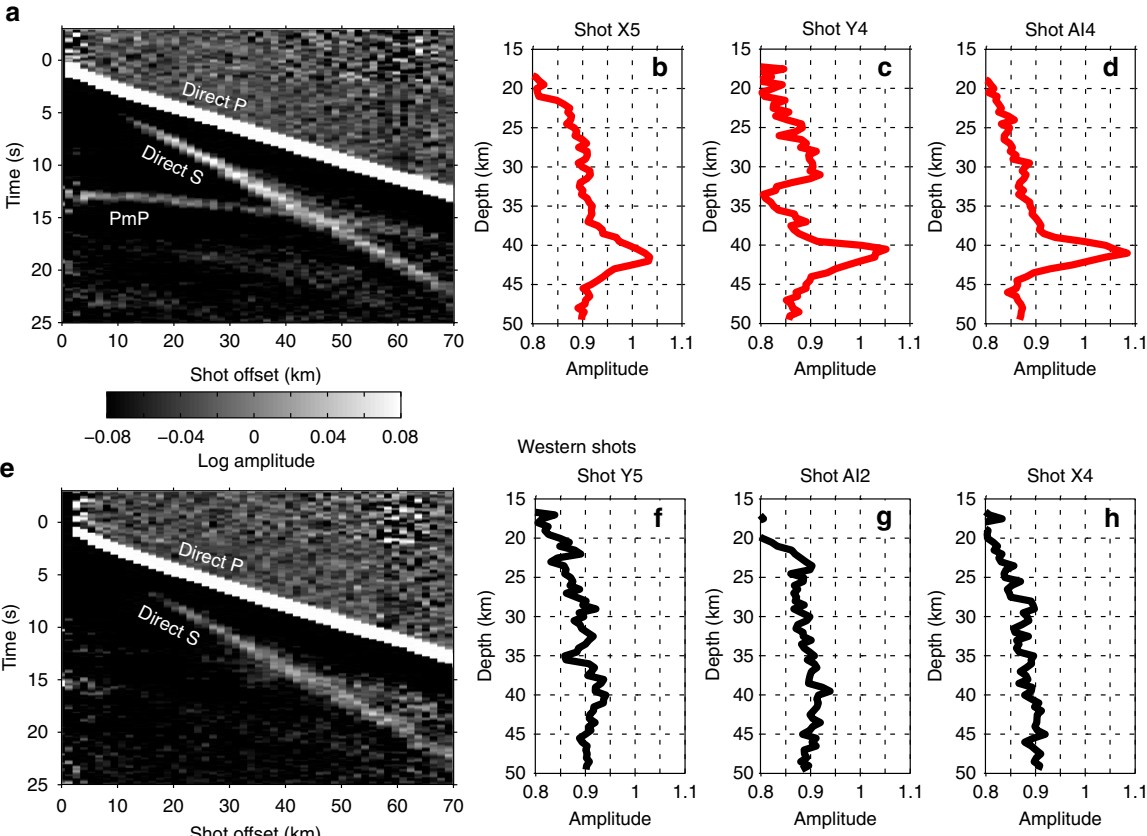

**Figure 3 | Stacked seismic traces showing the PmP amplitude variations.** Raw seismograms are bandpass filtered from 10 to 25 Hz and the short-term average to long-term average method has been applied. Plots along the top row (**a–d**) are from shots occurring on the east side of the volcano and plots on the bottom (**e–h**) are from the western shots. The images on the left (**a,e**) result from the binning and stacking of all traces by offset for each of the two shot subsets. Traces on the right (**b–d,f–h**) result from stacking all of the traces for an individual shot (see methods).

mantle velocity are primarily responsible for the observed reduction in Moho reflectivity across Mount St Helens.

## Discussion

Several previous active and passive-source seismic studies have observed weak or absent Moho arrivals at other locations within the Cascadia forearc region[14,16,18] (Fig. 1), including a reversal of the Moho velocity contrast in central Oregon[13]. The inferred reduction of mantle velocities has been attributed to the formation of low-velocity serpentine in the mantle wedge[11–14]. Serpentinite also has a relatively low-density and high-magnetic susceptibility, which is consistent with regional magnetic

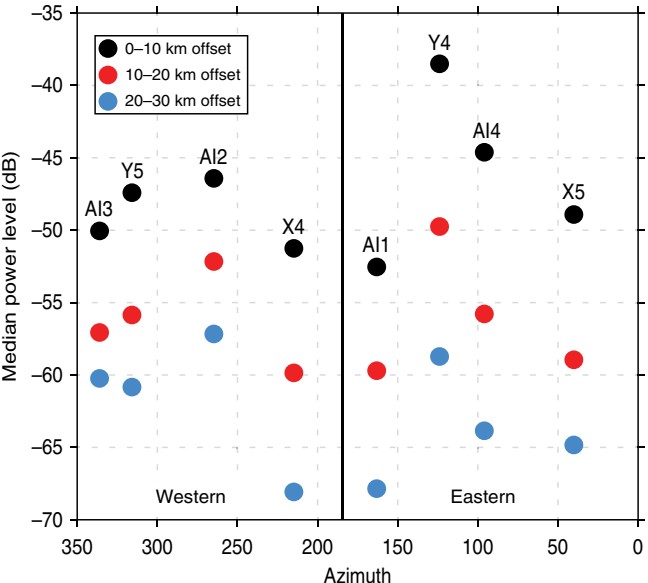

**Figure 4 | Comparisons of signal power for the eastern and western shots.** For these plots, the power spectrum is calculated for seismic traces recorded by the nodal array and the resulting spectral values are binned by frequency and shot offset. The inner ring shots (Fig. 2) are arranged by azimuth relative to Mount St Helens and the median power values are reported for the 10-25 Hz frequency band. With the exception of shot Y4, the recorded signal power is similar between the eastern and western shots.

and gravity anomalies[15]. The presence of ~50–75% serpentine is enough to reduce upper mantle P-wave velocities to 7.1–7.5 km s$^{-1}$ (ref. 12), similar to the observed lower crustal velocities beneath Mount St Helens[9,17]. However, we note that this is only a rough estimate of the amount of serpentine required to extinguish the Moho discontinuity and is on high end of reported estimates for Cascadia which range 15–60% (refs 13,19).

The high pressure serpentine phase (antigorite) is stable in the mantle wedge at relatively low temperatures (< ~700 °C (ref. 20)) and can form by hydrous alteration of peridotite[11,12]. From a global perspective, Cascadia is a relatively warm arc due to the young age of the subducting Juan de Fuca plate[21]. However, modelling suggests that the down-going slab dewaters primarily beneath the forearc region in warm subduction zones[22] and these fluids may therefore contribute to the serpentinization of the forearc mantle wedge[12]. Thermal modelling of the Cascadia subduction system[21] (Fig. 6) suggests that the eastern limit of the antigorite stability field could extend to near Mount St Helens, therefore, we interpret the lack of PmP arrivals west of Mount St Helens as evidence for a serpentinized upper mantle. The observed transition in Moho reflectivity is abrupt, occurring over a distance of about 5–10 km and is consistent with imaging the antigorite dehydration front whose rapid reaction kinetics[23] should produce a thermally well-defined boundary.

Areas where serpentinite has been inferred in the Cascadia forearc mantle are typically offset from the main axis of arc volcanism by ~50 km trenchward and are not usually associated with significant volcanism (Fig. 1). This observation is consistent with the view that the location of the volcanic arc relative to the trench is predominately controlled by the thermal structure of the mantle wedge[6,8]. Mount St Helens therefore presents a thermal paradox because it lies directly adjacent to the cold mantle wedge and yet still produces arc derived magmatism which requires elevated temperatures. For example, Mount St Helens primarily erupts dacites that are produced in the lower crust by either partial melting of a mafic source[24] and/or partial crystallization of mantle derived basalts[25]. Analysis of melt inclusions from the 1980 and 2008 eruption cycles indicate that magma temperatures were 860–900 °C (ref. 26), consistent with thermal models of arc lower crust (800–1,000 °C)[5]. Additionally, analysis of young (post-Miocene) mantle derived basalts from southern Washington suggest melt segregation temperatures that range ~1,200–1,450 °C (ref. 27). A hot mantle wedge directly

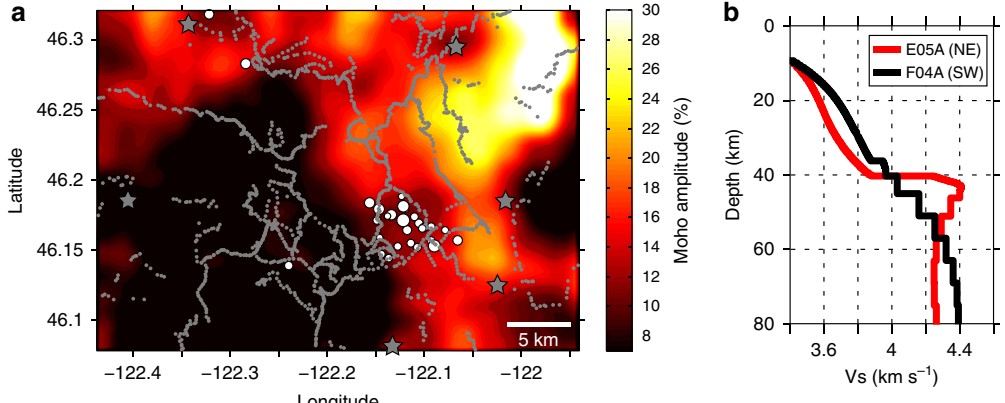

**Figure 5 | Spatial variations in the Moho reflectivity.** Moho amplitudes derived from common midpoint stacking of the PmP phase are shown in (**a**). Grey dots denote station locations and stars are the shot locations. The locations of deep long period earthquakes from the Pacific Northwest Seismic Network catalogue are plotted as white dots whose size scales with magnitude. Reported hypocenter uncertainties are all less than 6 km, however, mislocations and/or misidentifications are possible due to the emergent low frequency arrivals[34]. S-wave velocity models[16] for the two Transportable Array stations northeast (E05A) and southwest (F05A) of the image region are shown on in (**b**) (see Fig. 2 for their locations).

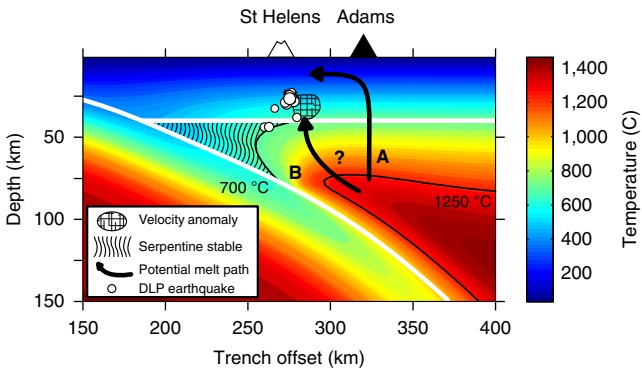

**Figure 6 | Two dimensional thermal model of the Cascadia subduction system in Central Oregon and geologic interpretation.** The surface locations of Mount St Helens and Mount Adams relative to the thermal model[21] are determined by using their distance from the subduction trench. The top of the subducting plate and the crust-mantle boundary of the overriding plate are plotted as white lines. The hachured area denotes the region of the mantle wedge where serpentine is stable ($< \sim 700 \,°C$ (ref. 20)). Locations of deep long period (DLP) earthquakes are plotted as white dots whose size scales with magnitude. The low velocity anomaly observed in the lower crust from iMUSH tomography[9] is shown. Arrows show two possible pathways for the lateral migration of melt towards Mount St Helens; These include a path through the mantle wedge (B) and one through the mid-crust (A).

beneath Mount St Helens is incongruent with the presence of a cold ($< \sim 700 \,°C$ (ref. 20)) serpentinized uppermost mantle just west of the volcano (Fig. 6).

One way this dilemma can be resolved is if the lower crustal source region resides east of the volcano, towards the hotter axial region of the volcanic arc, and the ascending melts migrate laterally in the crust towards the forearc region. Magnetotelluric results indicate that Mount St Helens is electrically connected to a region of high conductivity in the mid-crust[28], which is located near the southern terminus of the Southern Washington Cascades Conductor[29] (Fig. 1). This feature extends east to Mount Adams and has been interpreted as a layer of partial melt[28], which could thus provide a pathway for the westward migration of melts derived from the lower crust towards the cold forearc (see path A in Fig. 6). However, dacites erupted at Mount St Helens have notably different geochemistry than those found at Mount Adams which suggests that the melt source regions for these two volcanos are not the same[30].

Alternatively, recent iMUSH active-source tomography resolves a vertical column of low seismic velocities southeast of the volcano that extends from Moho to mid-crust depths and is interpreted as partial melt[9]. This feature could therefore represent a localized lower crustal source region for the volcanism at Mount St Helens. The mantle derived basalts that ultimately drive the magmatic system[25,31] require a hot source region ($> \sim 1,200 \,°C$ (ref. 27)) suggesting that these melts are formed further to the east and migrate west into the cold forearc mantle wedge (path B in Fig. 6). The true thermal state of the mantle wedge beneath Mount St Helens is undoubtedly complicated by effects not accounted for in the two-dimensional (2D) model (Fig. 6), such as three-dimensional (3D) topography of the slab surface[32] (Fig. 1) and heat transfer caused by the migration of melts[33]. Future geophysical constraints from the iMUSH project and increasingly realistic 3D modelling efforts are required to better constrain the deep melt pathways beneath Mount St Helens.

A cluster of deep long period (DLP) earthquakes has been identified just southeast of Mount St Helens in the lower crust (23–44 km depth, Fig. 6). DLP events are commonly observed in volcanically active regions, including beneath many of the

volcanoes in the Cascade Range[34], and these events are thought to be caused by the movement or cooling of magmatic fluids[35,36]. The DLP at Mount St Helens are located near the edge of the low velocity column observed in the iMUSH tomography (Fig. 6) and have been interpreted as the result of magma injection[9]. Our PmP results suggest the DLP earthquake cluster is located above the antigorite stability boundary in the mantle wedge (Fig. 5) and therefore these features may be related. Buoyant ascent of water produced by serpentine dehydration has been suggested to explain non-volcanic DLP events observed in the forearc region of Central Oregon[37]. A local heat source is required to drive dehydration[37] and this could be caused by the advection of heat associated with the arrival of mantle melts near the Moho beneath Mount St Helens (path B, Fig. 6).

## Methods

**Trace processing.** The raw seismic traces recorded by the Nodal and Texan arrays for each shot are bandpass filtered from 10 to 25 Hz. The sliding short-term average to long-term average ratio (STA/LTA) method[38,39] is then applied to the filtered traces to enhance the onset of the recorded seismic phases relative to their coda. A short-term window length of 0.2 s and a precursory long-term window length of 1 s are used. The direct P and S wave phases are removed from the data by applying two 3 s duration mutes to each trace that were centred on the predicted phase arrival times. Additionally, only traces from stations within 45 km of the source location are used for processing and stacking to avoid the interference of the PmP phase by the direct S wave arrival (Fig. 3).

**Travel-time calculations.** All travel-times are calculated with the Fast Marching Method[40,41] using a velocity model derived from a 3D tomography model[3] embedded within a 1D reference model[42] (Supplementary Fig. 1). Reflection travel-times from each point within the 3D image volume are calculated for a particular source-receiver pair using two travel-time models. The first model is simulated for a point source positioned at the shot location and provides the travel times from the shot location to all of the model points. The second model has the source positioned at the station location and calculates the travel time from all model points to the station (by reciprocity). Reflection travel-times are obtained by summing these two models at each image point.

**Moho imaging.** The array stacked trace for each shot (Fig. 3) is calculated by mapping each of the recorded traces from time-to-depth using the time-depth curve extracted from the reflection travel-time model at the source-receiver midpoint location. The stacked trace results from calculating the median value across all traces at each model depth. The 3D common midpoint (CMP) image used to derive the Moho reflectivity map (Fig. 5) is constructed using a similar technique, but the depth domain traces are binned laterally according to the source-receiver midpoint location and only traces within the same bin are stacked at each depth. Image bins are 4.5 km wide and are sampled every 0.5 km resulting in an 89% spatial overlap between adjacent pixels. The location and lateral extent of the CMP image volume was chosen to coincide with the region of highest data-fold (Supplementary Fig. 2). The Moho amplitude map (Fig. 5) is calculated by taking the maximum CMP image value occurring between 35 and 45 km depth at each model offset and the raw amplitude map is laterally smoothed by convolution with a five-point Hanning window for plotting purposes. An example 2D CMP cross section is shown in Supplementary Fig. 3. The values of the Moho amplitude map (Fig. 5) have been shifted and scaled so that they should be interpreted as signal-to-noise ratio in excess of 0.9, in percent. A reference value of 0.9 is used because it represents the background noise level due to the decaying coda, for example see the amplitude of the individual shot stacks in Fig. 3.

**Data availability.** The continuous seismic data recorded by the nodal array are available through the IRIS Data Management Center (http://ds.iris.edu/ds/nodes/dmc/) and are identified by the network code 1D (http://ds.iris.edu/mda/1D). Data recorded by the Texan instruments are available on request.

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

## Acknowledgements

Ellen Syracuse provided the thermal model for the central Cascadia region. Weisen Shen provided the shear wave velocity models for the Transportable Array stations. The deployment and data processing of the Nodal array was supported by NSF grants 1445937 and 1520875. The active-source component of the iMUSH project was supported by NSF grant 1459047.

## Author contributions

S.M.H. performed the data analysis and deployed nodal sensors. B.S. planned and deployed nodal sensors. A.L and E.K lead the active-source experiment. J.E.V., G.A.A. and K.C.C. contributed to design and implementation of the iMUSH experiment. All authors contributed to the interpretation of the results and writing the manuscript.

## Additional information

**Competing financial interests:** The authors declare no competing financial interests.

