## [Peer Review File · Nature Communications]

Reviewer #1:

The manuscript describes results and interpretation of a high-resolution active-source seismic experiment around Mount St Helens. The authors observe a dramatic amplitude variation of PmP phase from west to east across Mount St. Helens, and argue that the absence of a strong Moho conversion southwest of Mount St. Helens is due to serpentinization in the cold forearc mantle wedge. The research topic is certainly of interest but I am concerned that there are not enough information (or strong evidence) to judge/support the results and corresponding interpretation based on their model observation.

Major comments:

- The Cascadia subduction system is a relatively warm one compared to other subduction zones. The authors have been using “cold” for the forearc mantle wedge in the manuscript, however, by looking at the thermal model in Figure 5, the temperature beneath Mount St Helens within the wedge is 800-1000 degree. Only the very tip end of the mantle wedge is about 500 degree. In my observation, the forearc mantle wedge is not as cold as other systems. I suggest that the authors discuss a bit more about the thermal structure for global subduction zones and discuss how the various thermal models may affect serpentinization (and percent of serpentinization).
- The manuscript is based on the lack of PmP phase west of Mount St. Helen. How about other phases, such as PmS, SmS? Do the authors see any other phase conversions from other boundaries, such as the slab interface? I would suggest the authors look at the phases in a more detail and explore their relationships with the major boundaries.
- In the abstract and the last two paragraphs, the authors discuss possible processes that result in the volcanism beneath Mount St. Helens. However, I would suggest the authors also discuss what mechanism(s) actually drives the melt migration? Are there any other evidences, such as petrology or numerical modeling to support if path A or path B should be a more reasonable one?
- Line 58: 7-7.5 km/s for lower crust. This is relatively higher than average low crust velocity. What is the velocity right beneath the Moho from the tomography model? Does the tomography model support their observations here?
- What caused serpentinization? This is a question that need to be asked and addressed. Is this related to shallow dehydration of the subduction? Also 50-75% serpentine (in Line 72) is inferred in the manuscript, but what is the implication? How much water required? Is this possible for Cascadia? As serpentinization is key in this work, the author should address the origin of serpentine.
- There are a few places that the authors use “uncertain”, such as the relationship between the shear zone and the serpentine boundary (Lines 90-92) and the source of mantle melts (Lines 118-119). I would like to see if the authors have any thoughts as these seem two very important

questions to address. In particular, the melt source will be critical to their interpretation.

- Questions related to Figure 4. Please mark the two station locations (F04A and E05A) on Figure 4. It seems to me that these two stations (Figure 1) are far outside of their study area, thus, I am not sure how the 1D velocity profile beneath these two stations will help their interpretation. Also, the V_p/V_s ratio need to be at least 1.9 if considering $V_p=7.1-7.5$ km/s and $V_s=3.6-3.8$ in the lower crust from the manuscript. This is not the average V_p/V_s ratio for lower crust. If thought about this?
- I am a bit confused by Figure S1. The key feature that the authors are discussing is about the Moho variation. But the ray tracing in Figure S1 is only for top 4-5 km depth. How the ray tracing modeling contributes to their results?
- In my opinion, Figure 5 can be improved in a few ways to support their observation and interpretation. For example, showing the tomographic model, the magnetotelluric model, and distribution of earthquakes. This will help us to understand how the PmP observation in the manuscript corresponds to distribution of serpentine.

Minor comments:

- Line 7: within the Cascade arc
- Lines 33-34: How many days/months is the iMUSH experiment? What are the inter-spacings between the Texan and Nodal geophones, respectively?
- The sentence between Line 39-40: The resulting dataset provides an opportunity for high-resolution seismic imaging of deep crustal structure beneath this active arc volcano. Is there any seismic velocity model from this project that supports the serpentinization interpretation by this manuscript?
- Figure 1: Locations of the major active Quaternary volcanoes
- Figure 1: What are the two white dots on the left map?
- Line 63: that are located ...
- Lines 64-66: What are the shear-wave velocities in the lower crust and uppermost mantle beneath the two TA stations?
- Lines 76-78: This sentence is awful. Please rewrite.
- Line 86: At what depth is serpentine dehydration referred?
- Trace processing in Methods part: Please try to use present tense when describing steps of data processing.

PEER REVIEW FILE

Reviewers' comments:

Reviewer #2 (Remarks to the Author):

This manuscript presents interesting results on subsurface structure, seismicity and volcanism and their interrelations below Mount St Helens from data collected in the iMUSH experiment. A previously reported, arc-wide absence of continental Moho is examined in local detail and shown to possess a sharp boundary with regular crustal structure to the east. This boundary sits ~10 km to the southwest of Mt St Helens and also coincides with the presence of enigmatic, deep, long-period volcanic earthquakes. The Moho absence and volcanic earthquakes have been previously tied to serpentinization in the forearc mantle with the earthquakes occurring at the landward limit of stability where dehydration is thought to take place. This paper raises the interesting question as to how Mt St Helens volcanism is enabled in a situation where local mantle temperatures are presumably well below the solidus and proffers 2 plausible explanations invoking lateral melt migration.

The bearing of the results on arc magmatism, subduction zone zone (de)-hydration processes and enigmatic seismicity as supported by a seismic reflection data set of exceptional coverage and quality warrant publication in Nature Communications. I don't have any significant criticisms, though I anticipate that a more detailed account of will eventually appear in a longer-length publication. I have the following minor comments:

1. The forearc serpentinization / DLPV events are taken by the authors, understandably, to pose a thermal paradox with the presence volcanism at Mt St Helens, and they focus their discussion on this point. Is there, however, a possible role that fluids generated by serpentine dehydration (and possibly responsive for DLPVs) might play in facilitating magma generation?
2. Line 31. Mount St Helens is referred to as "the only major stratovolcano in the Cascadia forearc". I think this passage should be reworded to avoid confusion since, as a major stratovolcano, Mt St Helens in a sense partly defines the arc.
3. The topographic color scheme in Figure 1a should be modified to avoid the appearance that the Willamette Valley is an inland sea.

Reviewer #3 (Remarks to the Author):

Review by Kelin Wang

I am glad to be able to review this outstanding contribution. It is a very well written paper reporting important scientific results. I have always been intrigued by the location of Mount St. Helens which looks to be an exception to our conceptual models that work quite well to explain the location of volcanic arcs in general. The findings reported in this paper clarify the matter and raise new questions about melt migration. I recommend that the paper be accepted after some minor revision. My following comments are indexed by line numbers of the submitted manuscript.

29-30. Reference 7 (England and Katz, 2010) does not seem to be particularly relevant here, because it argues that the key process controlling the location of arc volcanism is dry melting, not flux melting.

54. lateral -> east-to-west

80. Although the dividing boundary is not truly N-S, I think the use of the word "southwest" brings more confusion than clarity. I suggest using "west".

82. Given the N-S variations, a distance of about 5 km may be a bit too precise.

82 and 86. Normally the cold forearc mantle wedge is an environment to form serpentinite. Serpentinite dehydration would require ongoing and perhaps local warming-up of the material, as proposed by Vidale et al. (2014). Without the specific warming mechanism explained, only those readers who have read and digested Vidale et al. (2014) can understand what you say here. To help most other readers, some explanation of the proposed warming process is needed here.

91. Fig. 3 -> Fig. 4

104-105. To the more general reader, the phrase "near adiabatic" does not follow the previous sentence very well; putting "hot" in front of "partially molten" is somewhat redundant. I suggest rephrasing the sentence as "A partially molten lower crust and a hot mantle wedge ..."

114. Fig. 4 -> Fig. 5

120. There is some problem in the sentence "This (melt migration) could be accomplished by ...". Melt migration is the cause of heat advection, not a result. Melts migrate to transport heat; "melt transport" would suggest the melts themselves are transported by something else. I suggest deleting it.

121. which -> that

Figures:

Fig. 1:

Shot locations AI1 and AI3 should be labelled here because they are used in Fig. 3. Perhaps label the white box with "Fig. 4"?

Caption should explain why some of the shot locations are labelled; define TA. Line 170: Fig. 3 -> Fig. 4.

Fig. 2:

To accentuate the lack of a clear Moho in the three western traces, I think it is important to use the same amplitude range (0.8 - 1.1) for their plots as for the eastern traces. If space is a problem, I suggest that the three traces be shown in the same plot using thinner lines (and do the same for the eastern traces).

In lines 176-177 of the caption, it will be helpful to the more general reader if a note is appended to each sentence in layman's terms, such as "as if ..."

Fig. 3:

The relative positions of AI4 and Y4 are reversed from those in Fig. 1. There must be an error in one of the two figures.

Fig. 4:

Line 193: Add "(E05A)" after "northeast" and "(F04A)" after "southwest".

Fig. 5:

The back arc area is too cold as shown (see Currie et al., 2004 EPSL). Wada and Wang (2009) got around dealing with the back arc by having the landward boundary rather close to the arc with assigned temperatures representing the thermal state of the back arc. Since the misleading back-arc thermal state shown in this figure is not needed by this paper, I suggest removing that part of the figure.

In line 201 of the caption, delete "the bottom of the overriding plate" and only say "the Moho", because the Moho is not the bottom of the overriding plate.

The authors' response to reviewers' comments.

To the editor and reviewers,

We believe that we have faithfully addressed the reviewers' concerns regarding the manuscript titled 'Seismic Evidence for a Cold Serpentinized Mantle Wedge Beneath Mount St Helens'. In particular we have added ~600 words and 6 references to the text, primarily to the introduction and discussion sections. This text is meant to provide some additional background material and to clarify the main conclusions of this work.

The individual comments of the reviewers are each addressed below and specific modifications made to the manuscript are reference were appropriate. Note, that the figure numbering has changed because we decided to split figure 1 into two separate figures. All line numbers and figure references refer to the edited manuscript.

Thank you all for your time and effort.

Reviewer #1 (from the PDF file)

The manuscript describes results and interpretation of a high-resolution active source seismic experiment around Mount St Helens. The authors observe a dramatic amplitude variation of PmP phase from west to east across Mount St. Helens, and argue that the absence of a strong Moho conversion southwest of Mount St. Helens is due to serpentinization in the cold forearc mantle wedge. The research topic is certainly of interest but I am concerned that there are not enough information (or strong evidence) to judge/support the results and corresponding interpretation based on their model observation.

Major comments:

- The Cascadia subduction system is a relatively warm one compared to other subduction zones. The authors have been using "cold" for the forearc mantle wedge in the manuscript, however, by looking at the thermal model in Figure 5, the temperature beneath Mount St Helens within the wedge is 800-1000 degree. Only the very tip end of the mantle wedge is about 500 degree. In my observation, the forearc mantle wedge is not as cold as other systems. I suggest that the authors discuss a bit more about the thermal structure for global subduction zones and discuss how the various thermal models may affect serpentinization (and percent of serpentinization).

Cascadia is indeed a warmer subduction zone when compared globally due to the relatively young age of the subducting Juan de Fuca slab. Our main conclusion is not that the Cascade arc is cold per se, but rather, that Mt. St Helens is located above the cold 'nose' of the mantle wedge. This result is surprising because the current thinking is that the location of the volcanic arc is controlled by the temperature of the mantle wedge and melting requires significantly higher temperatures.

As the reviewer correctly points out, the region where serpentinite is stable in the thermal model is an eastward narrowing area that hugs the Moho. Below this region in the center of the wedge temperatures are warmer, in the 800-1000C range, and could therefore exceed the saturated peridotite

solidus. However, as these melts ascend into the colder region of the wedge they could drop below the solidus temperature and solidify, thus preventing forearc volcanism (see Fig. 6, melt path A, in Grove et al., 2012). Although, whether or not ascending melt remains in thermal equilibrium with the solid matrix depends on the mode and geometry of melt transport (e.g. Hall and Kincaid, 2001 *Science*).

We also note that the estimated melt segregation temperatures for basalts erupted near the Washington Oregon border range ~1200-1450°C (Leeman et al., 2005), presumably some of which are being captured and drive volcanism at Mount St Helens (Hildreth, 2007). These temperatures are clearly too high to have been formed in the forearc wedge region given the thermal model (Figure 6). Finally, it should be stated that the thermal model is used only as a check to see if it is plausible that the 700°C isotherm could reach Mount St Helens. The true thermal structure in this area is surely more complicated.

To address the reviewers concern we have expanded the discussion section (lines 109-114) to include more background material, including the global context of the Cascadia arc and the influence of warm subduction on slab dewatering (added van Keken reference). We have also highlighted the shortcomings of the thermal model and the need for additional research efforts (lines 148-152).

- The manuscript is based on the lack of PmP phase west of Mount St. Helen. How about other phases, such as PmS, SmS? Do the authors see any other phase conversions from other boundaries, such as the slab interface? I would suggest the authors look at the phases in a more detail and explore their relationships with the major boundaries.

It is important to note that these sensors only measure the vertical component of ground motion and so it is difficult to detect PmS or SmS because these phases should have near vertical incidence at the shot offsets considered here. Because S-waves are transversely polarized, the dominant ground motion for these phases is in the horizontal plane and would therefore be difficult to detect with our sensors.

The shear wave velocity models in figure 5 are based upon surface wave and receiver function data. The receiver functions constrain the timing and amplitude of the Pms phase, and these data are the main constraint for the amplitude of the Moho velocity step in the models (see, Shen et al). The fact that there is no Moho velocity step for TA station F04A is a direct result of there being little or no Pms energy in the receiver function data, consistent with our PmP observations. This point has been clarified the main text (Lines 91-93).

For the imaging method we used in this paper, there are no other significant boundaries observed in the image volume (e.g. figure S2). While developing this approach, we also attempted to locate the slab surface by increasing the model volume to 125 km depth. However, no additional boundaries were observed. Also note that the shot and offset stacks in Figure 3 show little or no evidence of any other major scattered phases within the coda of the direct arrival (which is why we focused on PmP).

- In the abstract and the last two paragraphs, the authors discuss possible processes that result in the volcanism beneath Mount St. Helens. However, I would suggest the authors also discuss what mechanism(s) actually drives the melt migration? Are there any other evidences, such as petrology or numerical modeling to support if path A or path B should be a more reasonable one?

It remains unclear what mechanisms are responsible for the hypothesized melt migration. One of the goals of this paper is to highlight the challenge that Mount St Helens poses to models of arc processes and we hope that our results will motivate future studies focusing on these issues. To our knowledge, there are no modern numerical models specifically addressing melt migration in the context of forearc volcanism. Geochemistry and petrology methods are capable of constraining pressure (depth) and temperature conditions of melt, however, it is difficult to constrain the lateral movement of melt and distinguish different melt paths. However, there are differences between the geochemistry of the dacites erupted at Mount St Helens and Mount Adams which suggests that they tap different source regions. This observation would seem to support path B.

We have added this last point to the main text (Lines 141-142) and included the Defant and Drummond reference.

- Line 58: 7-7.5 km/s for lower crust. This is relatively higher than average low crust velocity. What is the velocity right beneath the Moho from the tomography model? Does the tomography model support their observations here?

The tomography model of Kiser et al. only uses seismic phases which propagate through the crust and therefore their model has no sensitivity to upper-mantle velocities. There is also a passive source seismic component to the iMUSH experiment, which is still ongoing, and will be able to better resolve the mantle velocity structure (lines 151-152). These data will be able to directly test the interpretation put forward in this paper, but for now, results from the much coarser transportable array (e.g. Shen et al.) are the best we have in this region.

- What caused serpentinization? This is a question that needs to be asked and addressed. Is this related to shallow dehydration of the subduction? Also 50-75% serpentine (in Line 72) is inferred in the manuscript, but what is the implication? How much water required? Is this possible for Cascadia? As serpentinization is key in this work, the author should address the origin of serpentine.

We have expanded our discussion of serpentine to include more background material (lines 109-114). This includes the formation of serpentine and its context within the Cascade arc. We have also added a comparison between our 50-75% estimate and values reported in the literature (lines 107-108). Calculating a water budget is a difficult task with many uncertainties (see the review of Reynard, 2013) and is outside the scope of the present study.

- There are a few places that the authors use “uncertain”, such as the relationship between the shear zone and the serpentine boundary (Lines 90-92) and the source of mantle melts (Lines 118-119). I would like to see if the authors have any thoughts as these seem two very important questions to address. In particular, the melt source will be critical to their interpretation.

Our data do not directly constrain the potential melt source regions and so the use of 'uncertain' was surely warranted here. We do suggest that the source region lies to the east and this melt path is shown in figure 6 (path B). We speculate about the melt source region and transport pathways but these remain open questions, ones that will be better resolved in future geophysical studies, such as the magnetotellurics component of iMUSH. This section has been rewritten for clarity and the term 'uncertain' has been removed (lines 143-152).

We have decided to remove the discussion pertaining to the Mount St Helens shear zone. The correlation with the location of the serpentine boundary is quite interesting but no concrete conclusions can be made and this discussion is tangential to the main focus of the manuscript.

- Questions related to Figure 4. Please mark the two station locations (F04A and E05A) on Figure 4. It seems to me that these two stations (Figure 1) are far outside of their study area, thus, I am not sure how the 1D velocity profile beneath these two stations will help their interpretation. Also, the V_p/V_s ratio needs to be at least 1.9 if considering $V_p=7.1-7.5$ km/s and $V_s=3.6-3.8$ in the lower crust from the manuscript. This is not the average V_p/V_s ratio for lower crust. If thought about this?

The two Transportable Array (TA) stations lie ~15-30 km outside of the image volume used in figure 5 and are difficult to plot on this figure due to its scale, however, their locations are plotted in Figure 2 (right) and this is referenced in the figure caption (lines 346-347). It would be nice if these stations were directly within the image volume but we are limited by the coarse sampling of the TA (~70 km station spacing). We also note that the iMUSH experiment will provide higher resolution passive-source data once the broadband deployment ends this summer.

The 1D results from these stations provide a direct and critical constraint to the interpretation of these data. Our results indicate that the Moho velocity step decreases to the west, however, our data do not directly constrain the cause of the reduced velocity step. These models are used to argue that the weak PmP phase observed to the west is caused by a reduction in upper-mantle velocities and reduced mantle velocities are interpreted to be caused by the presence of serpentine. We have rewritten the last two paragraphs in the results section and have attempted to clarify these points (lines 77-98).

Although these stations might be outside of the image volume, the passive-source data used to construct the velocity models is of much lower frequency (<1Hz) than the active-source data used in this study and the models should be thought of as a spatial average around the station location.

Furthermore, previous seismic observations suggest that the weak Moho region of the Cascadia forearc is likely a continuous feature observed over 100s of km along strike (our Figure 1, and Shen et al. Fig. 10b). Within this context, we believe that it is entirely reasonable to use the result from these two TA stations for interpreting our results.

Taking the ratio of lower crustal P-wave velocities derived from a 2D refraction model and S-wave velocities from a 1D inversion of passive data is not a good way to estimate crustal V_p/V_s . That being said, the high velocity lower crust observed within the Kiser and Parson models is interpreted as a mafic lithology, such as a mafic cumulate or gabbro, both of which are expected to have a relatively high V_p/V_s ratio at lower crustal conditions (~1.8-1.85 @ 0.4 GPa, see Christensen, 1996).

- I am a bit confused by Figure S1. The key feature that the authors are discussing is about the Moho variation. But the ray tracing in Figure S1 is only for top 4-5 km depth. How the ray tracing modeling contributes to their results?

This figure is meant to be an example of the travel-time calculations and pertains to the methods section. The first component shown in this figure is the velocity model used to calculate the travel time fields. The 3D velocity model (Waite and Moran, 2009) only extends to ~10 km depth, below that we use a 1D model and this is not shown. The second component of this figure are the ray paths produced by the Fast Marching code which is used to calculate the travel-time fields. We limited this plot to 4-5 km depth so that the velocity model could be seen but travel time fields are calculated for the full model volume (to 60 km depth). We added this point to the figure caption (lines 363-365).

- In my opinion, Figure 5 can be improved in a few ways to support their observation and interpretation. For example, showing the tomographic model, the magnetotelluric model, and distribution of earthquakes. This will help us to understand how the PmP observation in the manuscript corresponds to distribution of serpentine.

This is a good suggestion. We have added the distribution of DLP earthquakes and the region of low seismic velocity from the Kiser et al. study to this figure.

Minor comments:

- Line 7: within the Cascade arc

Corrected.

- Lines 33-34: How many days/months is the iMUSH experiment? What are the inter-spacings between the Texan and Nodal geophones, respectively?

The average instrument spacing (~250 m) has been added (lines 54-55).

We have opted not to delve into the details of the iMUSH schedule. There are several different components to this experiment and they all have different time requirements. For example, the passive-source seismic deployment will finish its two year recording period this summer.

- The sentence between Line 39-40: The resulting dataset provides an opportunity for high-resolution seismic imaging of deep crustal structure beneath this active arc volcano. Is there any seismic velocity model from this project that supports the serpentinization interpretation by this manuscript?

The short answer is, not yet (lines 49-50). As mentioned above, the iMUSH experiment is still ongoing and our final results are forthcoming. We have published 2D tomography results (Kiser et al.) but a full 3D model is not yet complete.

- Figure 1: Locations of the major active Quaternary volcanoes

Added 'Quaternary'.

- Figure 1: What are the two white dots on the left map?

Portland and Seattle, they have been removed from this plot.

- Line 63: that are located ...

Past tense is correct here. These stations are no longer recording.

- Lines 64-66: What are the shear-wave velocities in the lower crust and uppermost mantle beneath the two TA stations?

We have expanded this discussion and included the relevant numbers (lines 93-94).

- Lines 76-78: This sentence is awful. Please rewrite.

This sentence (and section) has been rewritten.

- Line 86: At what depth is serpentine dehydration referred?

Near the crust-mantle in the uppermost mantle, this sentence (and section) has been rewritten (lines 153-163).

- Trace processing in Methods part: Please try to use present tense when describing steps of data processing.

This has been corrected.

Reviewer #2 (Remarks to the Author):

This manuscript presents interesting results on subsurface structure, seismicity and volcanism and their interrelations below Mount St Helens from data collected in the iMUSH experiment. A previously reported, arc-wide absence of continental Moho is examined in local detail and shown to possess a sharp boundary with regular crustal structure to the east. This boundary sits ~10 km to the southwest of Mt St Helens and also coincides with the presence of enigmatic, deep, long-period volcanic earthquakes. The Moho absence and volcanic earthquakes have been previously tied to serpentinization in the forearc mantle with the earthquakes occurring at the landward limit of stability where dehydration is thought to take place. This paper raises the interesting question as to how Mt St Helens volcanism is enabled in a situation where local mantle temperatures are presumably well below the solidus and proffers 2 plausible explanations invoking lateral melt migration.

The bearing of the results on arc magmatism, subduction zone zone (de)-hydration processes and enigmatic seismicity as supported by a seismic reflection data set of exceptional coverage and quality warrant publication in Nature Communications. I don't have any significant criticisms, though I anticipate that a more detailed account of will eventually appear in a longer-length publication. I have the following minor comments:

1. The forearc serpentinization / DLPV events are taken by the authors, understandably, to pose a thermal paradox with the presence volcanism at Mt St Helens, and they focus their discussion on this point. Is there, however, a possible role that fluids generated by serpentine dehydration (and possibly responsive for DLPVs) might play in facilitating magma generation?

Serpentine dehydration requires a heat source and may therefore be an effect related to magma injection rather than a cause of magmatism. The formation of serpentine already requires the presence of water and so it is unclear whether or not the water associated with dehydration would cause any additional hydrous melting. Serpentine dehydration will affect the geochemistry of the arc melts (see Deschamps et al., 2013 *Lithos*) but this source has not currently been recognized at Mount St Helens. Additionally, distinguishing a shallow serpentinized source from other parts of the subduction system that may contain serpentine, such as in the down-going slab (e.g. van Keken et al., 2011) or along the slab interface (e.g. Reynard, 2015), would be difficult.

The paragraph discussing serpentine dehydration and its potential links to magma migration and the DLP earthquakes has been rewritten (lines 153-163).

2. Line 31. Mount St Helens is referred to as "the only major stratovolcano in the Cascadia forearc". I think this passage should be reworded to avoid confusion since, as a major stratovolcano, Mt St Helens in a sense partly defines the arc.

This paragraph of the manuscript (lines 36-46) has been rewritten for clarity and the offending text has been removed. We struggled with geographic description of Mount St Helens location but this is due to the fact that the Cascade arc violates many of the concepts of a 'normal' arc (Hildreth, 2007 has an excellent discussion on this).

3. The topographic color scheme in Figure 1a should be modified to avoid the appearance that the Willamette Valley is an inland sea.

This error in Figure 1 has been corrected.

Reviewer #3 (Remarks to the Author):

Review by Kelin Wang

I am glad to be able to review this outstanding contribution. It is a very well written paper reporting important scientific results. I have always been intrigued by the location of Mount St. Helens which looks to be an exception to our conceptual models that work quite well to explain the location of volcanic arcs in general. The findings reported in this paper clarify the matter and raise new questions about melt migration. I recommend that the paper be accepted after some minor revision. My following comments are indexed by line numbers of the submitted manuscript.

29-30. Reference 7 (England and Katz, 2010) does not seem to be particularly relevant here, because it argues that the key process controlling the location of arc volcanism is dry melting, not flux melting.

This reference has been removed.

54. lateral -> east-to-west

This correction has been made.

80. Although the dividing boundary is not truly N-S, I think the use of the word "southwest" brings more confusion than clarity. I suggest using "west".

We agree, corrected.

82. Given the N-S variations, a distance of about 5 km may be a bit too precise.

We have expanded the estimated distance to '5-10 km' (line 117).

82 and 86. Normally the cold forearc mantle wedge is an environment to form serpentine. Serpentine dehydration would require ongoing and perhaps local warming-up of the material, as proposed by Vidale et al. (2014). Without the specific warming mechanism explained, only those readers who have read and digested Vidale et al. (2014) can understand what you say here. To help most other readers, some explanation of the proposed warming process is needed here.

This is a good point, we have rewritten this paragraph and moved it to the end of the discussion section because it supports the path B interpretation (lines 153-163).

91. Fig. 3 -> Fig. 4

This correction has been made.

104-105. To the more general reader, the phrase "near adiabatic" does not follow the previous sentence very well; putting "hot" in front of "partially molten" is somewhat redundant. I suggest rephrasing the sentence as "A partially molten lower crust and a hot mantle wedge ..."

The suggested correction has been made.

114. Fig. 4 -> Fig. 5

Corrected.

120. There is some problem in the sentence "This (melt migration) could be accomplished by ...". Melt

migration is the cause of heat advection, not a result. Melts migrate to transport heat; "melt transport" would suggest the melts themselves are transported by something else. I suggest deleting it.

This sentence has been removed. The point we were trying to make is now on lines 148-151.

121. which -> that

Corrected.

Figures:

Fig. 1:

Shot locations AI1 and AI3 should be labelled here because they are used in Fig. 3. Perhaps label the white box with "Fig. 4"?

Caption should explain why some of the shot locations are labelled; define TA. Line 170: Fig. 3 -> Fig. 4.

Labels have been added for AI1 and AI3. The reason for labeling certain shots has been explained in the caption (lines 324). We opted not to label the white box directly in the figure to avoid clutter, however it is discussed in the caption. This is now Figure 2.

Fig. 2:

To accentuate the lack of a clear Moho in the three western traces, I think it is important to use the same amplitude range (0.8 - 1.1) for their plots as for the eastern traces. If space is a problem, I suggest that the three traces be shown in the same plot using thinner lines (and do the same for the eastern traces).

In lines 176-177 of the caption, it will be helpful to the more general reader if a note is appended to each sentence in layman's terms, such as "as if ..."

We have increased the range of axes so that all of the trace plots are now consistent.

We have removed the technical term 'move-out corrected', the reference to the methods section should be sufficient here.

Fig. 3:

The relative positions of AI4 and Y4 are reversed from those in Fig. 1. There must be an error in one of the two figures.

You are correct, Fig. 1 was in error and these labels have been corrected (now figure 2). We have also flipped the x-axis in figure 4 so that west is to the left and east is to the right.

Fig. 4:

Line 193: Add "(E05A)" after "northeast" and "(F04A)" after "southwest".

This text has been added and the directions SW and NW have been added to the figure legend.

Fig. 5:

The back arc area is too cold as shown (see Currie et al., 2004 EPSL). Wada and Wang (2009) got around dealing with the back arc by having the landward boundary rather close to the arc with assigned temperatures representing the thermal state of the back arc. Since the misleading back-arc thermal state shown in this figure is not needed by this paper, I suggest removing that part of the figure. In line 201 of the caption, delete "the bottom of the overriding plate" and only say "the Moho", because the Moho is not the bottom of the overriding plate.

We have cropped the rightmost 75 km from this figure to address the reviewer's concern.

This is a good point, 'Bottom of the overriding plate' has been changed to 'crust-mantle boundary of the overriding plate' in the caption (lines 352).

REVIEWERS' COMMENTS:

Reviewer #1 (Remarks to the Author):

The authors have addressed all my comments/questions/concerns. I suggest the manuscript to be published in the current format.

Reviewer #2 (Remarks to the Author):

The authors have adequately addressed my concerns and, as before, I regard this work as a valuable and important contribution worthy of publication in Nature Communications.

Note typo:

Line 120: "sepentinite" should be "serpentinite".

Reviewer #3 (Remarks to the Author):

The revised version and the rebuttal letter have fully addressed all my concerns and, as far as I can tell, carefully addressed the concerns of the other reviewers. I suggest that the revised version be accepted as is.

REVIEWERS' COMMENTS:

Reviewer #1 (Remarks to the Author):

The authors have addressed all my comments/questions/concerns. I suggest the manuscript to be published in the current format.

Reviewer #2 (Remarks to the Author):

The authors have adequately addressed my concerns and, as before, I regard this work as a valuable and important contribution worthy of publication in Nature Communications.

Note typo:

Line 120: "sepentinite" should be "serpentinite".

This typo has been corrected (Line 151).

Reviewer #3 (Remarks to the Author):

The revised version and the rebuttal letter have fully addressed all my concerns and, as far as I can tell, carefully addressed the concerns of the other reviewers. I suggest that the revised version be accepted as is.

We would like to thank our reviewers for their helpful comments which have greatly improved the final manuscript.